# Effects of Nutraceutical Compositions Containing Rhizoma Gastrodiae or Lipoic Acid in an In Vitro Induced Neuropathic Pain Model

**DOI:** 10.3390/ijms25042376

**Published:** 2024-02-17

**Authors:** Sara Ferrari, Simone Mulè, Rebecca Galla, Arianna Brovero, Giulia Genovese, Claudio Molinari, Francesca Uberti

**Affiliations:** 1Laboratory of Physiology, Department of Translational Medicine, University of Piemonte Orientale, Via Solaroli 17, 28100 Novara, Italy; sara.ferrari@uniupo.it (S.F.); simone.mule@uniupo.it (S.M.); rebecca.galla@uniupo.it (R.G.); genogiuly@hotmail.it (G.G.); 2Noivita Srls, Spin-Off, Via Alfieri 3, 28100 Novara, Italy; 3Dipartimento di Scienze Cliniche e Biologiche, Università Degli Studi di Torino, 10043 Torino, Italy; arianna.brovero@unito.it; 4Dipartimento per lo Sviluppo Sostenibile e la Transizione Ecologica, University of Piemonte Orientale, 13100 Vercelli, Italy; claudio.molinari@uniupo.it

**Keywords:** gastrodin, intestinal absorption, neuropathic pain model, cell pain signalling, anti-inflammatory properties

## Abstract

Background: Peripheral neuropathy is caused by a malfunction in the axons and myelin sheaths of peripheral nerves and motor and sensory neurons. In this context, nonpharmacological treatments with antioxidant potential have attracted much attention due to the issues that some conventional pharmaceutical therapy can generate. Most of these treatments contain lipoic acid, but issues have emerged regarding its use. Considering this, the present study evaluated the beneficial effects of nutraceuticals based on *Gastrodiae elata* dry extract 10:1 or lipoic acid in combination with other substances (such as citicholine, B vitamins, and acetyl L-carnitine). Method: To assess the combination’s absorption and biodistribution and exclude cytotoxicity, its bioavailability was first examined in a 3D intestinal barrier model that replicated oral ingestion. Subsequently, a 3D model of nerve tissue was constructed to investigate the impacts of the new combination on the significant pathways dysregulated in peripheral neuropathy. Results: Our findings show that the novel combination outperformed in initial pain relief response and in recovering the mechanism of nerve healing following Schwann cell injury by successfully crossing the gut barrier and reaching the target site. Conclusion: This article describes a potential alternative nutraceutical approach supporting the effectiveness of combinations with *Gastrodiae elata* extract in decreasing neuropathy and regulating pain pathways.

## 1. Introduction

Pain is a frequent sign of many illnesses and typically denotes the presence of tissue damage; in contrast, ‘chronic pain’ is the term used to describe pain that lasts for extended periods, either as a result of a disease process or after the typical healing time for an accident [1]. Approximately one in nine young adults experience chronic pain worldwide, corresponding to 11.6% [2]. This increased pain has drastic and expensive effects on the population regarding work performance and quality of daily life. As a result, there is a need for new and more effective treatments, and the discovery of novel mechanisms and pathways offers crucial direction for drug discovery initiatives [3]. While initial studies on pain mechanisms focused on the role of neurons of the central nervous system (CNS), researchers’ attention has now shifted to the role of peripheral neurons. In this context, peripheral neuropathy (PN) is a disorder caused by the malfunctioning of peripheral nerves, including sensory and motor neurons, as well as their axons and myelin sheaths. Severe discomfort, muscular weakness, and sensory loss are clinical symptoms of PN [4]. The fundamental causes of prolonged pain are neuronal hyperactivity and/or hyperexcitability, which arise following nerve injury. Among the ideas that explain the mechanisms associated with central neuropathy include intracellular signalling, microglial activation, neurotrophic factors, elevated levels of pro-inflammatory and inflammatory mediators, and oxidative stress (OS) [5]. Indeed, increasing research suggests that OS has a role in neuropathic pain via peripheral and central sensitisation. Excess reactive oxygen and nitrogen species (ROS, RNS) harm neurons by interacting with proteins, carbohydrates, lipids, and nucleic acids. In this context, SIRT3 has an important role in several biological processes, including mitochondrial function [6]. Furthermore, OS causes inflammatory cells, including lymphocytes, monocytes, neutrophils, basophils, and eosinophils, to produce a range of substances (proinflammatory cytokines, vasoactive amines and peptides, eicosanoids). It has been established that neuropathic pain is caused by the overexpression of proinflammatory mediators (TNFα, IL-1) [7]. In this context, current medications still have difficulty treating this pathological pain, which is a maladaptive mechanism. Due to this, nutraceuticals (dietary supplements and herbal/natural products) are becoming increasingly popular due to their low cost, high nutritional and therapeutic qualities, and capacity to bind to many molecular targets [8]. In recent years, lipoic acid (LA) has emerged as a viable alternative; its mechanisms of action in experimental diabetic neuropathy include reduced OS, enhanced neural blood flow, nerve conduction velocity, and a variety of other nerve function indicators [9]. At present, it is known that the phytoprotection system has collected a large number of complaints of adverse responses to dietary supplements based on LA; among them are cases of autoimmune hyperinsulinemia. This uncommon illness, also known as Hirata syndrome, is a severe adverse response characterised by extreme hypoglycaemia [10]. Otherwise, the adverse effects of LA are correlated with diabetic polyneuropathy, which also includes gastrointestinal symptoms with treatment at higher doses (>600 die) [11]. Therefore, scientific research is looking for alternatives to counter the negative effects of this substance. In this regard, *Gastrodia elata* Blume (*G. elata*) is a plant traditionally used to treat various conditions such as headaches, dizziness, spasms, epilepsy, stroke, amnesia, and other disorders in Eastern countries [12]. The part of this plant that has medicinal action is its rhizome (*Rhizoma gastrodiae*), from which over 200 bioactive components and secondary plant metabolites have been isolated, including Gastrodin, gastrodigenin (p-hydroxybenzyl alcohol), polysaccharides (PEG), vanillin, parishin, trace elements, amino acids, organic acids, and other compounds [13]. Among these, Gastrodin (glycoside known as 4-hydroxy benzyl alcohol-4-O-β-D-glucopyranoside) [14,15] is the main active constituent, which is considered a phytochemical marker of *G. elata* extract used to control the quality of the extract, and this is the reason to find the *G. elata* extract named gastrodin [13].

Gastrodin has been extensively investigated for its biological actions; as a result, numerous pharmacological activities have been attributed to Gastrodin, including anti-inflammatory, antioxidant, sedative, hypnotic, anti-vertigo, analgesic, anti-epileptic, antidepressant, and anxiolytic activities [16]. Gastrodin’s pharmacokinetics have been studied in several models, and in vitro studies indicate that Gastrodin is mostly transported via a passive paracellular transport system [17]. Because glucose transporters (GLTs) were shown to be involved in the intestinal absorption of Gastrodin, glucose and GLT inhibitors might influence the process [18]. Furthermore, the primary metabolite of Gastrodin is p-hydroxy benzyl alcohol (HBA) and following oral dosing, Gastrodin is rapidly converted into HBA in the intestines, plasma, kidney, liver, and brain [19]. It is also worthy of interest that acute toxicity studies indicate that Gastrodin and its metabolite HBA are safe when administered at high doses. Indeed, oral administration of Gastrodin or HBA in mice at doses up to 5000 mg/kg resulted in no mortality or harmful consequences. Although toxicity studies showed that Gastrodin is largely safe to use, occurrences of clinical adverse drug responses (ADRs) or events (ADEs) caused by Gastrodin have been documented on occasion [20]. However, in neuropathic pain, *G. elata* has been demonstrated to possess anti-inflammatory and neuroprotective effects in vivo [21].

In this context, innovative formulations with anti-inflammatory effects addressing this disease condition are crucial today. Dietary supplements can benefit neuropathy patients by delivering a robust natural antioxidant capable of combatting free radicals and regenerating various vitamins. In this context, citicholine (Cit), which is globally available as a dietary supplement, has comprehensive neuroprotective properties and beneficial effects on neurodegenerative diseases, raising the level of serotonin [22], decreasing glutamate levels, responsible for damage to the brain during ischemia [23], and synthesising the phosphatidylcholine, which may stimulate the repair and regeneration of damaged cell membranes of neurons [22]. Moreover, long-term citicoline therapy enhanced the endogenous neurogenesis processes in patients with stroke or depression [22]. Furthermore, another endogenous molecule with antioxidant properties is acetyl-L-carnitine (ALC), which acts on neurotrophic factors like nerve growth factor (NGF), affects brain neurotransmitters such as acetylcholine, serotonin, and dopamine, and protects against OS. Long-term neurotrophic and analgesic efficacy has also been observed in animal models of neuropathic and chronic inflammatory pain [24]. Generally, it is acknowledged that ALC plays a neurotrophic role through the modulation of neurotrophic factors such as brain-derived neurotrophic factor (BDNF) and NGF [24]. Due to these characteristics, ALC can treat neuropathic pain by significantly improving mean nerve conduction velocity and amplitude in the sensory and motor nerves, reducing pain and relapse. Meanwhile, it is also widely known that peripheral neuropathy can result from a B vitamin deficiency; indeed, B vitamins have historically been used as treatments in cases of vitamin deficiency neuropathy. Regarding B6, neuropathy can result from both excess and insufficiency. At the same time, B12 supplementation seems to lessen the signs and symptoms of diabetic neuropathy [25]. In injured neurons, vitamin B12 reduces OS, aberrant cell signalling, and glutamate-induced neurotoxicity [26]. Several in vitro and in vivo investigations have demonstrated that B vitamins decrease mechanical allodynia and have antinociceptive, antihyperalgesic, and anti-inflammatory properties [27,28]. Similarly, it has been reported that B vitamins can enhance analgesics’ therapeutic benefits [29,30]. The B vitamins are important participants that preserve the vitality of neurons in various ways, shielding nerves from harmful external stimuli. Specifically, vitamin B6 regulates neuron metabolism, vitamin B12 preserves myelin sheaths, and vitamin B1 functions as a site-directed antioxidant. These vitamins facilitate the formation of new cell structures and significant regeneration. Indeed, the lack of these vitamins will promote chronic discomfort and nerve degradation, ultimately resulting in peripheral neuropathy [31].

Thus, considering that various processes contribute to the pathophysiology of pain, it may be more effective to utilise a combination of multiple agents, each with a unique mode of action, to reduce pain simultaneously at lower doses and with fewer side effects. The present study aims to replace this element in the presence of *G. elata* combined with Cit, ALC, and the vitamin B class to improve their antioxidant and anti-inflammatory properties to reduce pain signalling in an in vitro model.

## 2. Results

### 2.1. Dose–Response Study of the Single Components on Caco-2

Before exploring the potential beneficial effects against neuropathic pain of the new formulation, the effects of the single agents were evaluated on the viability of the Caco-2 cell line using a dose–response study to verify the toxicological analysis, considering a period between 2 h and 6 h as the relevant exposure time. As shown in Figure 1A, all substances improved CaCo-2 cell viability, as reported in the literature, compared to the control (*p* < 0.05). These effects are more relevant considering *G. elata* 100 µg/mL, since it improved mitochondrial metabolism by about 40% compared to the control (*p* < 0.05), indicating a better profile in terms of cell viability compared to the other concentrations tested (*p* < 0.05). In addition, regarding Cit and ALC, although they had a similar time course profile, Cit 250 mM and ALC 7.5 µM demonstrated a better trend compared to the other concentrations (*p* < 0.05). Finally, vitamin B group (Vit. B group) 1 mg/mL exerted the most beneficial effects compared to 100 µg/mL and 200 µg/mL (*p* < 0.05). For this reason, the concentration able to maintain or produce little increase in cell viability (*p* < 0.05 vs. control) was chosen for each agent to be used in the final formulation. In particular, *G. elata* 100 µg/mL, Cit 250 mM, ALC 7.5 µM, and Vit. B group 1 mg/mL were chosen and used on all successive experiments.

Indeed, when all the agents were added together (Figure 2A), significant changes were observed (*p* < 0.05), indicating the maintenance of a self-protection response even by the complete formulation (named Assonal). Additionally, Assonal exerted a better performance compared to other substances tested (*p* < 0.05), particularly compared to the commercial product (containing 50 µM LA, 250 mM Cit, 7.5 µM ALC, and 1 mg/mL Vit. B group), which was used as a positive control (*p* < 0.05). This may be due to *G. elata*’s greater effectiveness over LA (*p* < 0.05), which enhances mitochondrial metabolism and produces a more favourable response. The same results were also found by analysing the ROS production (Figure 2B) since Assonal exerted a significant effect compared to the other substances tested, including the commercial product (*p* < 0.05).

### 2.2. Permeability and Absorption Mechanisms Analyzed in an In Vitro Intestinal Barrier Model

Based on the results obtained in this study’s first phase, additional investigations were performed to learn key information on intestine absorption and transport mechanisms, by creating an in vitro intestinal barrier model. In this context, the transepithelial electrical resistance (TEER), the intestinal absorption, and the TJ, such as Occludin, Claudin-1, and ZO-1, were evaluated. As reported in Figure 3A, TEER analysis showed that all tested agents actively support optimal absorption and maintain correct intestinal homeostasis (*p* < 0.05). Specifically, the passage through the intestinal epithelium demonstrates that *G. elata* 100 µg/mL maintained the epithelial integrity, increasing the ionic flux of the paracellular exchanges across the intestinal epithelial better than the other single agents (*p* < 0.05). The enhancing effect of *G. elata* 100 µg/mL is particularly appreciated when combined with Cit 250 mM, ALC 7.5 µM, and Vit. B group 1 mg/mL. Indeed, Assonal showed more beneficial effects than other commercial products (*p* < 0.05). These data were confirmed with the TJ analysis; as shown in Figure 3B–D, *G. elata* 100 µg/mL exerted the most significant effects compared to the other single agents for all the TJs tested (*p* < 0.05). Also, in this case, Assonal exerted the most significative effect compared to the commercial product regarding claudin-1 (about 51%, *p* < 0.05), occludin (about 47%, *p* < 0.05), and ZO-1 (about 57%, *p* < 0.05) and compared to control value (reported as 0-line, *p* < 0.05). Furthermore, these data are supported by *G. elata* 100 µg/mL alone, demonstrating ameliorative effects on TJ compared to LA 50µM (*p* < 0.05). Finally, data on absorption analysis (Figure 3E) supported the results reported above since *G. elata* 100 µg/mL on its own possessed a slightly higher bioavailability than LA 50 µ (*p* < 0.05), demonstrating the ability to cross the intestinal barrier in a particularly efficient way and allowing Assonal to have a higher absorption profile compared to the commercial product containing LA (*p* < 0.05).

### 2.3. Effects of Single Components and Combination on 3D EngNT Co-Cultures

Considering the data obtained by absorption analysis, further experiments were conducted on 3D-engineered neural tissue (3D EngNT) to investigate the ability of *G. elata* 100 µg/mL, Cit 250 mM, ALC 7.5 µM, and Vit. B group 1 mg/mL, alone and combined, to reach and act on the peripheral nerve after intestinal passage in physiological conditions. All data were compared to the commercial product and its main ingredient. As shown in Figure 4, all the single agents tested could reach the target site after intestinal passage without negatively affecting mitochondrial metabolism or the inflammatory response (*p* < 0.05). In particular, *G. elata* 100 µg/mL was able to amplify the cell viability compared to the other single agents (*p* < 0.05, Figure 4A), demonstrating an ability to maintain mitochondrial well-being and, at the same time, showing an ability to reduce the amount of inflammatory marker (*p* < 0.05). *G. elata* 100 µg/mL also exerted a significant effect compared to LA 50 µM on cell viability and TNFα production (approximately 44% and 33%, respectively, *p* < 0.05). Moreover, Assonal induced a significative effect compared to the commercial product in both the parameters tested (about 15% for cell viability and 24% for TNFα production, *p* < 0.05, respectively), demonstrating the higher proficiency of this combination. NGF levels were analysed to confirm their effects on the physiological conditions promoting neuronal well-being. As reported in Figure 4C, in this case, *G. elata* 100 µg/mL exerted the most beneficial effect compared to the single agents (*p* < 0.05), demonstrating this substance’s effectiveness over raw materials already available on the market as well (*p* < 0.05). Moreover, this advantage was confirmed by its combination with the other substances: Assonal induced a significant impact compared to commercial products (about 42% *p* < 0.05). Finally, the expression of pERK/MAPK was also analysed considering that, following nerve injury, phosphorylation of ERK increases. Indeed, it is known that nerve injury-induced phosphorylation of ERK occurs early and is long-lasting, so the regulation of pERK/MAPK may be considered a promising therapeutic target for treating neuropathic pain [32]. As expected, the effect of *G. elata* 100 µg/mL was greater than LA 50 µM (about 16%, *p* < 0.05); as a final confirmation of the effects observed so far, also in this case, the greatest effect was obtained following treatment with Assonal (*p* < 0.05).

### 2.4. Effects of Single Components and Combination on an In Vitro Model of Peripheral Nervous Injury (PNI)

To replicate strong demyelination before stimulation with the two formulations and the single drugs, the 3D EngNT was pre-treated with 200 ng/mL glial growth factor 2 (GGF) starting on day 14 of maturation. This was necessary since the study needed to replicate the peripheral nerve tissue damage. Therefore, further studies were conducted to investigate the effects on cell viability and ROS production (Figure 5A,B). As expected, nerve tissue treated with only 200 ng/mL GGF produced considerable ROS and significantly less biological activity (*p* < 0.05). In contrast, both individual components and formulations could not only re-establish physiological conditions but induced a significant increase in cell viability and a decrease in ROS production (*p* < 0.05). Indeed, Assonal increased cell viability by about 19% and decreased ROS production almost four times compared to commercial products (*p* < 0.05). Additionally, regarding the inflammatory panel, even if all the agents were able to reduce the production of TNFα and IL-2 compared to GGF 200 ng/mL alone (*p* < 0.05), once again, Assonal demonstrated more significant effects even compared to commercial products, with a decrease in TNFα production of 3 times and IL-2 production of 2.7 times (*p* < 0.05). This is due to the combination based on *G. elata* 100 µg/mL, which, thanks to its powerful antioxidant effect, can reverse the damaging effects.

Additionally, considering that Nav1.7 and Nav1.8 are highly expressed sodium channels in pain pathophysiology, further analysis was performed to investigate them. Particularly, sodium channels are naturally implicated in transmitting information from pain receptors in the peripheral nervous system (PNS) to the CNS [33]. As reported in Figure 6 (panels A,B), all the tested agents were able to induce a decrease in both Nav1.7 and Nav1.8 proteins compared to GGF 200 ng/mL (*p* < 0.05). Unsurprisingly, even in this instance, the main effects were performed by the novel formulation Assonal, which induced a decrease in the NaV1.7 protein level of almost 2.5 times compared to the commercial product and a reduction in the NaV1.8 protein level of nearly 10 times compared to the commercial product (*p* < 0.05). Specifically, these data demonstrate how Assonal can reduce pain by reducing the level of NaV1.7 and NaV1.8 channels, which are deputed to modulate pain. Several studies look at the function of GABA receptors near sodium channels, which facilitate spike propagation in myelinated axons (Figure 6C). Notably, the analysis of GABA level, whose functionality was statistically increased by the treatment with all agents alone compared to GGF 200 ng/mL (*p* < 0.05), further supported the beneficial effects of *G. elata* 100 µg/mL. Indeed, aided by its presence, Assonal decreased GABA-A more than the commercial product (approximately one time more, *p* < 0.05).

Lastly, the in vitro modulation of specific molecular pathways involved in neuropathic pain was investigated. As shown in Figure 7A, *G. elata* 100 µg/mL is efficient in repristinating the myelinisation following nerve injury caused by 200 ng/mL GGF (*p* < 0.05), as demonstrated by p75 analysis and, consequently, in the inhibition of MPZ protein. Specifically, Assonal was able to completely revert the damage induced by GGF 200 ng/mL (about 77% more, *p* < 0.05) while also causing an increase in p75 expression compared to the commercial product (about 59%, *p* < 0.05). Moreover, the level of MPZ, another marker involved in maintaining the myelin sheath, was improved following the administration of *G. elata* 100 µg/mL, Cit 250 mM, ALC 7.5 µM, and Vit. B group 1 mg/mL alone and combined with Assonal (*p* < 0.05). Under these circumstances, Figure 7B illustrates the beneficial activity of *G. elata* 100 µg/mL under GGF 200 ng/mL condition (*p* < 0.05), and subsequently, the impact of Assonal on both it and commercial product (about 1.5 times and 70%, respectively, *p* < 0.05).

Further evidence of the amelioration of the nerve injury and restoration of the myelinisation process was given by neuregulin 1 (NRG1) and estrogen receptor beta (ERb) analyses (Figure 7C,D). In detail, regarding NRG1 level, LA 1 mg/mL was not able to improve it compared to GGF 200 ng/mL (*p* < 0.05), while its combination in the commercial product formulation achieved this result; in contrast, both *G. elata* 100 µg/mL and, particularly, Assonal reverted GGF 200 ng/mL-induced damages, improving NRG1 levels (about 18% and 31%, respectively, *p* < 0.05). Furthermore, Assonal obtained a significant activity compared to the commercial product (about 58%, *p* < 0.05), demonstrating the higher efficacy of this novel formulation. Also, ERb analysis revealed a similar trend: the PNI condition impacted its activity, but the administration of all the agents tested announced a reversal and a restoration of this condition (*p* < 0.05). In particular, more evident effects were obtained with *G. elata* 100µg/mL and, particularly, with Assonal, and the latter exerted a significant influence compared to commercial products (about 33%, *p* < 0.05). The analysis of NGF levels gives the final piece of evidence that it restores the correct mechanism. Even in this case, *G. elata* 100 µg/mL alone exerted a higher effect than LA 50 µM (*p* < 0.05), and, when included in the novel combination, improved the self-restoration response. Indeed, the most significant impact was exerted by Assonal, with a substantial increase in NGF levels compared to commercial products (approximately 30%, *p* < 0.05).

## 3. Discussion

Numerous nonpharmacologic treatments, including non-invasive methods like exercise therapy, integrated cognitive behavioural therapy, and nutritional supplements, have been suggested to treat neuropathic pain [34]. Indeed, in recent years, there has been a lot of interest in so-called “nutraceuticals” and other nonpharmaceutical supplements, which may be able to complement current pharmaceutical-based treatment for chronic neuropathic pain [35]. For this reason, a possible new combination, called Assonal, has been investigated, as summarised in Appendix A.

### 3.1. Analysis of Assonal Effects on 3D In Vitro Intestinal Barrier Model

In this context, numerous nutraceuticals emerged recently to treat neuropathic pain in a variety of illnesses, including peripheral neuropathy, diabetic neuropathy, and neuropathic pain associated with chemotherapy [34]. The present investigation aimed to find an effective new food supplement that could aid PNI patients in managing their discomfort. Specifically, the effects of *G. elata* (called OXADIA) were investigated compared to LA and in combination with ALC, Cit, and vitamins of the B group to evaluate if their combination (called Assonal) could be more effective than a commercial product based on the same formulation but without *G. elata*, plus LA demonstrated optimal results regarding cell viability and less involvement of oxidative-inducing mechanisms compared with LA alone, a component of the original formulation of the commercial product. This allowed us to confirm the lowest irritant action at the intestinal level promoted by the nutraceutical component under investigation while reinforcing any complications reported from the oral use of LA [10]. This was also confirmed by the increased intestinal TJ levels, which are crucial parameters in evaluating an epithelial monolayer set-up in vitro. Indeed, analyses of Zo-1, which mediates adhesion, claudin-4, which maintains the structure, and occluding, which contributes to stabilisation, confirm proper gut function. In addition, TEER analysis confirms a more active role of *G. elata*, hypothesising better uptake compared to LA. All this information was confirmed by a good absorption rate obtained by correlating it with the results of the fluorescent probe used. Indeed, it demonstrated a better percentage of absorption rate than LA alone; similar results could also be found in the two compositions, with Assonal exerting a better effect than commercial products, suggesting a possible synergistic effect between the single components.

### 3.2. Analysis of Assonal Effects on a Physiological 3D EngNT Model in Physiological Conditions

After intestinal passage, all the nutraceutical substances under examination were characterised on the specific target of the study, a 3D nervous system. Specifically, the antioxidant and anti-inflammatory action described in the literature [21] were evaluated. Indeed, the production of TNFα, an inflammatory marker, was supplanted after stimulation with *G. elata*, indicating minor neuronal damage. Also, fundamental mechanisms in maintaining viability, functionality, and cell survival were examined; additionally, the NGF level was analysed. NGF was the first neurotrophin identified as having a key role in the survival and function of sensory and sympathetic neurons in the peripheral nervous system (PNS) and cholinergic neurons of the basal forebrain in the CNS [36]; specifically, it is a pleiotropic factor involved in growth, differentiation, and repair of neuronal damage. Therefore, its increased activity is directly related to better cellular viability. The pERK/MAPK evaluation has been used fundamentally since the pERK/MAPK pathway, traditionally associated with important roles in cell proliferation and differentiation, has strong relevance at the peripheral level in maintaining neuronal plasticity. In addition, pERK/MAPK expression has recently been shown to contribute to nociceptive responses that follow inflammation and/or nerve injury [31]. Indeed, the intracellular pathway pERK/MAPK was also evaluated, with *G. elata* exerting the most beneficial effect. Also, in this case, the superiority of Assonal compared to the commercial product was assessed.

### 3.3. Analysis of Assonal Effects on a 3D EngNT Model under PNI Conditions

As a result of these data in physiological nervous conditions, our research proceeded with the characterisation at the level of a 3D nervous model in the condition of damage (induced with 200 ng/mL GGF), resulting in demyelination. In this study phase, the findings largely confirmed the antioxidant (reduction of ROS levels) and anti-inflammatory (lowering TNFα and IL-2 productions) effects promoted by *G. elata*, which were amplified after treatment with Assonal due to the antioxidant and anti-inflammatory effects of the other components. Specifically, Assonal supplanted the inflammatory mechanism and OS. Furthermore, normal peripheral nerve growth, migration, and differentiation of Schwann cells are regulated by NRG1 activating ERb receptors; the condition of PNI modifies NRG1/ERb signalling by disturbing the balance between NRG1 isoforms and reducing the expression of multiple molecules involved in cell survival [37]. While the disruption of NRG-1/ERb B2 signalling and impaired neurotrophic support are undoubtedly linked to the degeneration of sensory neurons in PNI, most likely in the biological activity of Schwann cells, treatment with Assonal restored the altered neurotropism, preventing damage to motor fibres and slowing down nerve conduction, demonstrating its neurotropic properties.

In addition, it outlines the role of the proposed new formulation not only from an antioxidant and anti-inflammatory point of view but also in characterising its effect on mechanisms directly involved in initiating neuropathic pain. In the literature, the role of the sodium channels Nav1.7 and Nav1.8, widely expressed in the pathophysiological condition of pain, has been defined as crucial [33]. The analyses of their level confirmed the effects of the analysed substance, as they were all able to lessen the expression and levels of Nav1.7 and Nav1.8, especially after treatment with Assonal; the same results were also observed with GABA-A, an important inhibitory neurotransmitter in the mammalian CNS [38]. Furthermore, data revealed that Assonal appears to be able to repair the damage on the myelin sheath that protects the axon and simultaneously act on NGF release and bind the p75 neurotrophin receptor, reproducing the mechanism of analgesia observed in humans by mimicking the neuropathic pain condition in vitro, specifically inflammation and OS, leading to an imbalance of ion transport. Indeed, as previously shown [39], a rise in NGF levels in various inflammatory situations may be a key feature of the human chronic pain syndrome. As a result, lowering the inflammatory response was critical for mitigating the detrimental effects of chronic pain. Conversely, pain is associated with a functional imbalance between MPZ, ERb, and NRG1, whose relative balance is crucial for maintaining Schwann cell homeostasis during PNI. NRG1 normally activates ERB receptors in peripheral nerves to regulate Schwann cell growth, migration, differentiation, and dedifferentiation; however, PNI alters NRG1/ERb signalling by disrupting the balance between NRG1 isoforms, decreasing the expression of several molecules involved in cell survival, and thus activating the MAPK pathway [37]. While sensory neurodegeneration in PNI is associated with impaired neurotrophic support and disruption of NRG-1/ERb signalling, presumably in the biological activity of Schwann cells, Assonal treatment restored the altered neurotropism, preventing nerve conduction slowing and motor neuron damage. It is possible to assume that the new formulation has more beneficial effects due to the different mechanisms of action and targets of *G. elata* and LA. Specifically, LA is an antioxidant, free radical scavenger, and metal ion chelator that is synthesised in the mitochondria and contributes to the enzymatic breakdown of nutrients [40].

On the other hand, research has shown that *G. elata* possesses anti-inflammatory and antioxidant qualities due to its capacity to increase the Nrf2 nuclear translocation. Thus, the capacity to improve intracellular antioxidant responses is linked to the neuroprotective benefits of *G. elata*, since Nrf2-mediated antioxidant and detoxification protein production are important antioxidant responses to protect against ROS [41].

All these outcomes indicate the potential usability of a novel formulation composed of *G. elata* 100 µg/mL (OXADIA), Cit 250 mM, ALC 7.5 µM, and Vit. B group 1 mg/mL, capable of crossing biological membranes, notably the intestinal barrier, and reaching the target location, showing optimal effects in the in vitro model of PNI.

## 4. Materials and Methods

### 4.1. Agents Preparation

*Gastrodiae elata* blume (*G. elata*) dry extract 10:1 (called OXADIA^TM^) was extracted from *Rhizoma* and is characterised by a particle size of 100 mesh and the presence of Gastrodin. *G. elata*, ALC, and Vit. B group (composed of 22.5% vitamin B1 and B2, 16.2% vitamin B5, 9% vitamin B6, and 29.7% vitamin B12) were tested alone and combined to verify their effectiveness in crossing the intestinal barrier and reaching the PNS. Specifically, *G. elata* was tested in a range from 100 µg/mL to 300 µg/mL [42,43], Cit in a range from 100 mM to 300 mM [44], ALC in a range from 7.5µM to 30 µM [45] and Vit. B group from 100 µg/mL to 1 mg/mL [46,47]. All substances tested were prepared directly in Dulbecco’s Modified Eagle’s Medium (DMEM, Merck Life Science, Rome, Italy) without phenol red and supplemented with 0.5% foetal bovine serum (FBS; Merck Life Science, Rome, Italy), 2 mM L-glutamine (Merck Life Science, Rome, Italy), and 1% penicillin–streptomycin (Merck Life Science, Rome, Italy) for all analyses. Based on their beneficial effects, the best concentration of each substance was chosen and combined in a new formulation composed of *G. elata* 100 µg/mL, Cit 250 mM, ALC 7.5 µM, and Vit. B group 1 mg/mL (called Assonal). Moreover, Assonal was analysed throughout the project by comparing its effects on individual activities, alongside LA 50 µM alone and a commercial product (composed of LA, Cit, ALC, and Vit. B group, named Old Assonal). The LA was used at a concentration of 50 µM as reported in the literature [48], and the commercial product was tested at the same dosage as Assonal, containing the same percentage of Cit, ALC, and Vit. B group. Furthermore, all substances tested were prepared as already mentioned and donated by AGAVE S.r.l., which obtained them from Vivatis Pharma GBHE.

### 4.2. Cell Culture

#### 4.2.1. Epithelial Intestinal Cells

The human epithelial intestinal CaCo-2 cell line was used as a model to predict the features of human intestinal absorption following oral intake [49,50]. Cells were supplied with American Type Culture Collection (ATCC) and cultured in Dulbecco’s Modified Eagle’s Medium Advance (DMEM-Adv, Thermo Fisher Scientific, Rodano, MI, Italy) supplemented with 5% FBS, 2 mM L-glutamine, and 1% penicillin–streptomycin, maintained in an incubator at 37 °C and 5% CO_2_ and 95% of humidity. Experiments were performed using CaCo-2 cells at passage numbers between 26 and 32 to maintain the physiological balance among paracellular permeability and transport properties [51]. Cells were plated differently: 1 × 10^4^ cells on 96-well plates to study cell viability using an MTT-based In Vitro Toxicology Assay Kit (Merck Life Science, Rome, Italy) and ROS production, which were tested after synchronising cells for 8 h with DMEM without red phenol and supplemented with 0.5% FBS, 2 mM L-glutamine, and 1% penicillin–streptomycin maintained in an incubator at 37 °C with 5% CO_2_. In addition, 2 × 10^4^ cells were plated on 6.5 mm Transwell^®^ (Corning^®^ Costar^®^, Merck Life Science, Rome, Italy) with a 0.4 μm pore polycarbonate membrane insert (Corning^®^ Costar^®^, Merck Life Science, Rome, Italy) in a 24-well plate to perform the absorption analyses [52].

#### 4.2.2. Schwann Cell

A rat-derived Schwann, RSC96, cell line purchased from ATCC was cultured in DMEM-Adv (Thermo Fisher Scientific, Rodano, MI, Italy) supplemented with 5% FBS, 2 mM L-glutamine, and 1% penicillin–streptomycin [53] and maintained in an incubator at 37 °C, 5% CO_2_, and 95% humidity. Experiments were conducted using RSC96 cells between 10 and 15 passages, subculturing 2–3 times a week.

#### 4.2.3. Neuronal Cell

The rat neuronal PC12 cell line, supplied by ATCC, was cultured in Roswell Park Memorial Institute-1640 Advanced (RPMI-Adv, Thermo Fisher Scientific, Rodano, MI, Italy) supplemented with 2 mM glutamine, 5% horse serum (HS; Merck Life Science, Rome, Italy), and 5% FBS. The cultures were maintained at sub-confluency in an incubator at 37 °C, 5% CO_2_, and 95% humidity; cells used for experiments were between 3 and 13 passages [54]. PC12 cells are among the most eligible and frequently employed neuronal cell lines for in vitro screening for neuroprotective compounds [55]. A total of 4 × 10^6^ RSC96 cells and 1 × 10^5^ PC12 cells were plated in a co-culture system to reproduce the 3D EngNT in vitro, creating the peripheral nerve environment in vitro model [53].

### 4.3. Experimental Protocol

This study was carried out in three phases, and a schematic diagram of each phase of the experimental protocol is reported below (Figure 8). The starting point of the study protocol was intended to study the absorption and transport processes of the novel formulation and individual components through the intestinal barrier using a 3D in vitro model with Caco-2 cells [51]. The analyses included mitochondrial metabolism and cellular well-being, OS, permeability and measurement of transepithelial electrical resistance (TEER), absorption, and tight junction (TJ) activity. In the second phase, the product metabolised by the intestinal cells was subsequentially in contact with a 3D model for studying the peripheral nerve (3D EngNT), evaluating cell metabolism, inflammation, and some molecular pathways. Finally, in the third phase, peripheral nerve damage and demyelination were simulated in vitro, treating 3D EngNT with GGF 200 ng/mL, to explore cell survival and mitochondrial activity, OS and ROS quantification, inflammatory processes involved in neuropathy, nociceptive pathways for pain control, and the main pathways involved in the processes of myelin sheath protection and neurite formation [53].

### 4.4. Cell Viability

MTT-based In Vitro Toxicology Assay Kit (Merk Life Science, Rome, Italy) was performed on a 96-well plate to determine cell viability, following a classical protocol reported in the literature [51]. The absorbance of all solubilised samples (treated and untreated) was measured at 570 nm with correction at 690 nm by a spectrophotometer (Infinite 200 Pro MPlex, Tecan, Männedorf, Switzerland) and the results were expressed by comparing the data to the control sample (untreated samples defined as the 0% line) and reported as the means of five independent experiments performed in triplicate.

### 4.5. ROS Production

The ROS production was quantified by analysing the reduction of cytochrome C using a standard protocol [51]. The absorbance was measured at 550 nm through a spectrophotometer (Infinite 200 Pro MPlex, Tecan, Männedorf, Switzerland). O_2_ ratio was expressed as the mean ± SD (%) of nanomoles per reduced cytochrome C per microgram of protein compared to the control (untreated samples) of five independent experiments performed in triplicate.

### 4.6. In Vitro Intestinal Barrier Model

An in vitro intestinal barrier model was created, following a standard protocol reported in the literature, using the Transwell^®^ system to assess whether the substances used can cross the intestinal barrier [51,56] following the European Medicines Agency (EMA)- and Food and Drug Administration (FDA)-approved protocols [57,58], which are used to predict the absorption, metabolism, and bioavailability of several substances after oral intake in humans. Briefly, CaCo-2 cells, plated on a Transwell^®^ insert, were maintained in a complete medium changed every other day, both at the basolateral and apical part for 21 days before the stimulation [51]. The TEER values were assessed using EVOM3 and STX2 chopstick electrodes (World Precision Instruments, Sarasota, FL, USA) throughout the entire maturation time to evaluate mature intestinal epithelium formation and a proper paracellular mechanism. Absorption analysis started on the 21st day when TEER values were ≥400 Ωcm^2^ [59]. On the apical side, before the stimulation, the culture medium was verified to be pH 6.5, which resembles the pH in the lumen of the small intestine. In contrast, pH 7.4 on the basolateral side represented blood [51,60]. The cells were stimulated with all substances from 2 to 6 h before the successive analyses. At each time point, the uptake was detected using a fluorescent tracer at a concentration of 0.04% (Santa-Cruz, CA, USA) [61]. The amount of fluorescein carried was measured at 37 °C by incubating CaCo-2 cells for 40 min at the above concentration. Fluorescence was detected with a fluorescence spectrophotometer (multilabel plate reader, VICTOR X4, Perkin Elmer, Waltham, MA, USA) at excitation/emission wavelengths of 490/514 nm. Results are expressed as the proportion (%) of the original amount permeating the cells. The permeation rate [nmol min (mg protein)], J, was calculated as follows:J = Jmax [C]/(Kt + [C])
where

-Jmax: the maximum permeation rate;-[C]: the initial concentration of fluorescein;-Kt: the Michaelis–Menten constant.

Results are expressed as mean ± SD (%), and negative controls without cells were tested to exclude influence by the Transwell^®^ membrane.

### 4.7. TJs Analysis

The CaCo-2 lysates were used to analyse occludin in a Human Occludin (OCLN) ELISA Kit (MyBiosource, San Diego, CA, USA), claudin-1 in an ELISA Kit (Cusabio Technology LCC, Huston, Houston, TX, USA), and Zona Occludens 1 (ZO-1) in the human tight junction protein 1 (TJP1) ELISA kit (MyBiosource, San Diego, CA, USA) following the manufacturer’s instructions [51]. The absorbance was measured by a spectrophotometer (Infinite 200 Pro MPlex, Tecan, Männedorf, Switzerland) at 450 nm. The data were obtained by comparing them to the standard curve (from 0 to 1500 pg/mL for occludin, and from 0 to 1000 pg/mL for claudin-1 and ZO-1). They were expressed as a percentage (%) versus the control (0 line) of five independent experiments performed in triplicate.

### 4.8. 3D EngNT Co-Culture Setup

The interaction between RSC96 and PC12 cell lines is a key feature for mimicking the peripheral nerve environment in vitro, regenerating neurites, and supporting Schwann cells [53,62,63]. According to the literature, the 3D nerve tissue model was prepared [53]. Briefly, 1 mL of a solution containing 80% (*v*/*v*) Type I rat tail collagen (2 mg/mL in 0.6% acetic acid, Thermo Fischer, Milan, Italy), 10% (*v*/*v*) Minimum Essential Medium (MEM, Merck Life Science, Milano, Italy), 5.8% (*v*/*v*) neutralising solution (Biosystems, Monza, Italy), and 4.2% Schwann cell suspension (4 × 10^6^ RSC96 cells per 1 mL gel) was added to a rectangular scaffold with dimensions of 16.4 mm × 6.5 mm × 5 mm. When the gel had set, it was immersed in 10 mL DMEM and incubated at 37 °C with 5%, CO_2_ for 24 h to permit cellular self-alignment; at the end, the gel was stabilised using plastic compression (120 g weight for 1 min). Each stabilised aligned cellular gel was cut into equal segments according to the samples to be treated. Each gel segment was transferred to a 24-well plate, and 1 × 10^5^ PC12 was seeded on top of each segment for establishing the co-cultures; this passage is crucial to permit neurite extension across the horizontal plane following the aligned Schwann gels. The 24-well plate containing gels was maintained in an incubator at 37 °C with 5% CO_2_ and 95% humidity to allow attachment of neuronal cells to the collagen gel; next, 1 mL of culture medium (DMEM, Merck Life Science, Rome, Italy) supplemented with 10% (*v*/*v*) FBS, 100 U/mL of penicillin, and 100 μg/mL of streptomycin was added to each well.

### 4.9. TNFα ELISA Assay

The supernatant of 3D EngNT was used to quantify TNFα production using the TNFα ELISA kit (Merck Life Science, Rome, Italy) according to the manufacturer’s instructions [64]. The absorbance of the samples was measured at 450 nm using a plate reader (Infinite 200 ProMPlex, Tecan, Männedorf, Switzerland). The data were obtained and compared to the standard curve (ranging from 0 to 6000 pg/mL), and the results were expressed as a mean ± SD (%) versus the control (0 line) of five independent experiments performed in triplicate.

### 4.10. Human Beta-NGF Assay

The Rat beta-NGF ELISA kit (Abcam, Cambridge, UK) was used on the supernatant of 3D EngNT following the manufacturer’s instructions [65]. Briefly, 100 μL of diluted samples were incubated overnight at 4 °C and washed four times with 1× Wash Buffer, and then 100 μL of detection antibody was added to each well and incubated for 1 h at room temperature with gentle shaking. Then, the wells were washed four times, and 100 μL of streptavidin–HRP and the plate were incubated for 45 min. After the incubation, the wells were washed again, and 100 μL of TMB substrate was added. Finally, the plate was incubated for 30 min at room temperature in the dark with gentle shaking, and the reaction was stopped with 50 μL of Stop Solution. The absorbance was measured by the spectrometer at 450 nm (Infinite 200 Pro MPlex, Tecan, Männedorf, Switzerland) and expressed as pg/mL compared to a standard curve (ranging from 15 to 15,000 pg/mL). The results were reported as the mean ± SD (%) versus control of five independent experiments performed in triplicate.

### 4.11. ERK/MAPK Expression

Three-dimensional EngNT lysates were used to assess ERK/MAPK expression using the InstantOneTM ELISA (Thermo Fisher, Milan, Italy) following the manufacturer’s instructions [61]. To each well of the InstantOne ELISA microplate strips, 50 µL of each lysate was added. The antibody cocktail and microplate strips were then incubated for 1 h at room temperature on a microplate shaker. After 20 min of application of the detection reagent, the reaction was stopped by adding a Stop Solution. The Infinite 200 Pro MPlex (Tecan, Männedorf, Switzerland) spectrometer, which operates at a wavelength of 450 nm, was used to measure the strips. Average absorbance (%) compared to the control was used to represent the results.

### 4.12. Interleukin-2 ELISA Assay

Interleukin-2 quantification was determined using the Rat Interleukin-2 (IL-2) ELISA Kit (FineTest, Wuhan, China) according to the manufacturer’s instructions on 3D EngNT lysates [66]. The absorbance was read at 450 nm using a spectrophotometer (Infinite 200 ProMPlex, Tecan, Männedorf, Switzerland). A standard curve was plotted relating the intensity of the colour (OD) to the concentration of standards (ranging from 31.25 to 2000 pg/mL), and the results were expressed as mean ± SD (%) versus control (0 lines) of five independent experiments performed in triplicate.

### 4.13. NAV1.7 ELISA Assay

NAV1.7 quantification was determined using the SCN9A/Nav1.7 ELISA Kit (LifeSpan BioSciences, Lynnwood, WA, USA) on 3D EngNT lysates according to the manufacturer’s instructions. Briefly, 100 µL of each sample was incubated for 2 h at 37 °C in a pre-coated plate; after that, the samples were removed and 100 µL of Detection Reagent A was added for 1 h at 37 °C. After washing the plate 3 times, 100 µL of Detection Reagent B was added for 1 h at 37 °C. The plate was washed again 5 times before adding 90 µL of TMB Substrate. After 20 min at 37 °C, 50 µL of Stop Solution was added to each well, and the absorbance was read at 450 nm using a spectrophotometer (Infinite 200 ProMPlex, Tecan, Männedorf, Switzerland). A standard curve was plotted relating the intensity of the colour (OD) to the concentration of standards (ranging from 0.312 to 20 ng/mL), and the results were expressed as mean ± SD (%) versus control (0 lines) of five independent experiments performed in triplicate.

### 4.14. NAV1.8 ELISA Assay

NAV1.8 quantification was determined using the Mouse Sodium Channel Protein Type 10 Subunit Alpha (SCN10A) ELISA Kit (MyBiosource, San Diego, CA, USA) on 3D EngNT lysates according to the manufacturer’s instructions [67]. The absorbance was read at 450 nm using a spectrophotometer (Infinite 200 ProMPlex, Tecan, Männedorf, Switzerland). A standard curve was plotted relating the intensity of the colour (OD) to the concentration of standards, and the results were expressed as mean ± SD (%) versus control (0 lines) of five independent experiments performed in triplicate.

### 4.15. γ-Aminobutyric Acid (GABA) ELISA Assay

The Rat GABA (γ-aminobutyric acid) ELISA Kit (FineTest, Wuhan, China) was used on 3D EngNT lysates following the manufacturer’s instructions [68]. The absorbance was measured at 450 nm through a plate reader (Infinite 200 ProMPlex, Tecan, Männedorf, Switzerland). The data were obtained and compared to the standard curve (6 to 400 pg/mL). The results were expressed as mean ± SD (%) versus control (0 lines) of five independent experiments in triplicate.

### 4.16. p75 by NGFR ELISA Assay

According to the manufacturer’s instructions, the Rat NGFR ELISA kit (MyBiosource, San Diego, CA, USA) was used on 3D EngNT lysates [69]. The plate was read immediately at 450 nm using a spectrophotometer (Infinite 200 ProMPlex, Tecan, Männedorf, Switzerland). The data were obtained and compared to the standard curve (0.312 to 20 ng/mL). The results were expressed as a mean ± SD (%) versus control (0 lines) of five independent experiments performed in triplicate.

### 4.17. Myelin Protein Zero (MPZ) ELISA Assay

Per the manufacturer’s instructions, the MPZ level was determined using a Rat ELISA kit (My-BioSource, San Diego, CA, USA) in 3D EngNT lysates [69]. The plate was read at 450 nm using a spectrophotometer (Infinite 200 ProMPlex, Tecan, Männedorf, Switzerland). The concentration was expressed as ng/mL compared to a standard curve (ranging from 0.06 to 18 ng/mL), and the results were reported as the mean ± SD versus the control (0 lines) of five independent experiments performed in triplicate.

### 4.18. (Neuregulin 1)NRG1 ELISA Assay

According to the manufacturer’s instructions, the NRG1 Rat ELISA Kit (FineTest, Wuhan, China) was used on 3D EngNT supernatants [69]. The enzymatic reaction was measured by a spectrophotometer (Infinite 200 Pro MPlex, Tecan, Männedorf, Switzerland) at 450 nm. The results were obtained by comparing the data to the standard curve (ranging from 0.156 to 10 ng/mL). They were expressed as a percentage (%) versus the control (0 lines) of five independent experiments performed in triplicate.

### 4.19. Estrogen Receptor β ELISA Assay

The Rat Estrogen Receptor β (ERβ) ELISA Kit (Cloud-Clone, Houston, TX, USA) was used on 3D EngNT lysates, according to the manufacturer’s instructions [70]. The absorbance was measured by a spectrophotometer at 450 nm (Infinite 200 ProMPlex, Tecan, Männedorf, Switzerland) and expressed as pg/mL compared to a standard curve (ranging from 0.312 to 20 ng/mL). The results were reported as the mean ± SD (%) versus control of five independent experiments performed in triplicate.

### 4.20. Statistical Analysis

One-way analysis of variance (ANOVA) and Bonferroni post hoc tests were used to process the data acquired using Prism GraphPad statistical software 9.4.1. A Student’s t-test with two tails was adopted to compare the two groups. A two-way ANOVA was conducted to evaluate multiple group comparisons, followed by a two-sided Dunnett post hoc test. The mean ± SD of at least five independents performed in triplicate variables was used to express all results.

## 5. Conclusions

In conclusion, this study demonstrated how Assonal containing *G. elata* 100 µg/mL (OXADIA^TM^) Citicoline citicholine Cit 250 mM, ALC 7.5 μM, and Vit. B group 1 mg/mL has proven to be an adequate alternative to common analgesic drugs to relieve the mechanism underlying pain caused by PNI, as shown in Appendix A. Consequently, our findings support for the first time the concept that this novel formulation can efficiently reduce neuropathy by modifying the fundamental pain mechanism by stimulating the recovery mechanisms of PNS cells. Further, Assonal prevented motor fibre damage and the slowing down of nerve conduction by restoring the altered neurotropism. In conclusion, the current study gives new insights into the effects of nutraceuticals on neuropathic pain disorders thus Assonal can be regarded as a novel and safe neuropathy treatment.

## Figures and Tables

**Figure 1 ijms-25-02376-f001:**
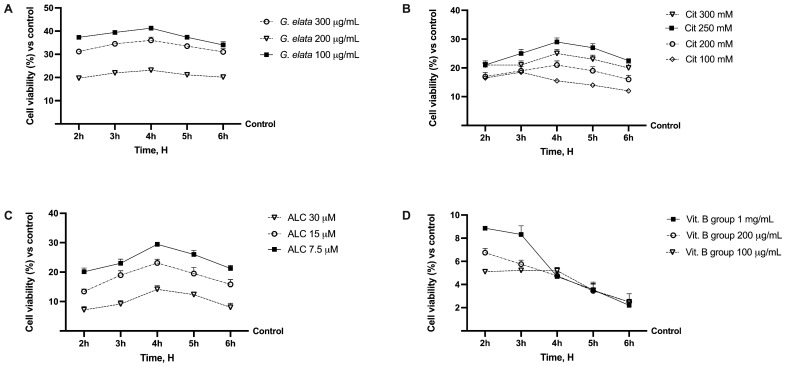
Analysis of different substances in dose–response and time-dependent (from 2 h to 6 h) studies evaluating cell viability by MTT test. In (**A**), *G. elata* = *Gastrodia elata* Blume 10:1; in (**B**), Cit = citicholine; in (**C**), ALC = acetyl-L-acetyl-L-carnitine; in (**D**), Vit. B group effects on cell viability. Data are expressed as mean ± SD (%) of 5 independent experiments normalised to the control (0%) line.

**Figure 2 ijms-25-02376-f002:**
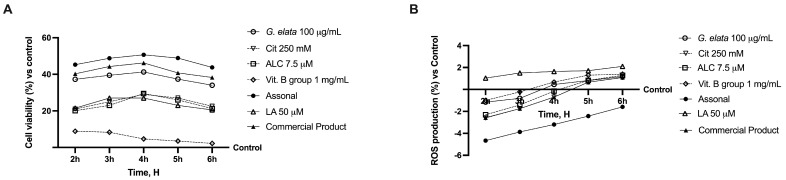
Analysis of the cell viability (**A**) and ROS production (**B**) of the single components and their combinations in time-dependent study (from 2 h to 6 h). The abbreviations are the same as reported in Figure 1. LA = 50 µM; Assonal = 100 µg/mL *G. elata* plus 250 mM Cit plus 7.5 µM ALC plus 1 mg/mL Vit. B group; commercial product = 50 µM LA plus 250 mM Cit plus 7.5 µM ALC plus 1 mg/mL Vit. B group. Data are expressed as mean ± SD (%) of 5 independent experiments normalised to control.

**Figure 3 ijms-25-02376-f003:**
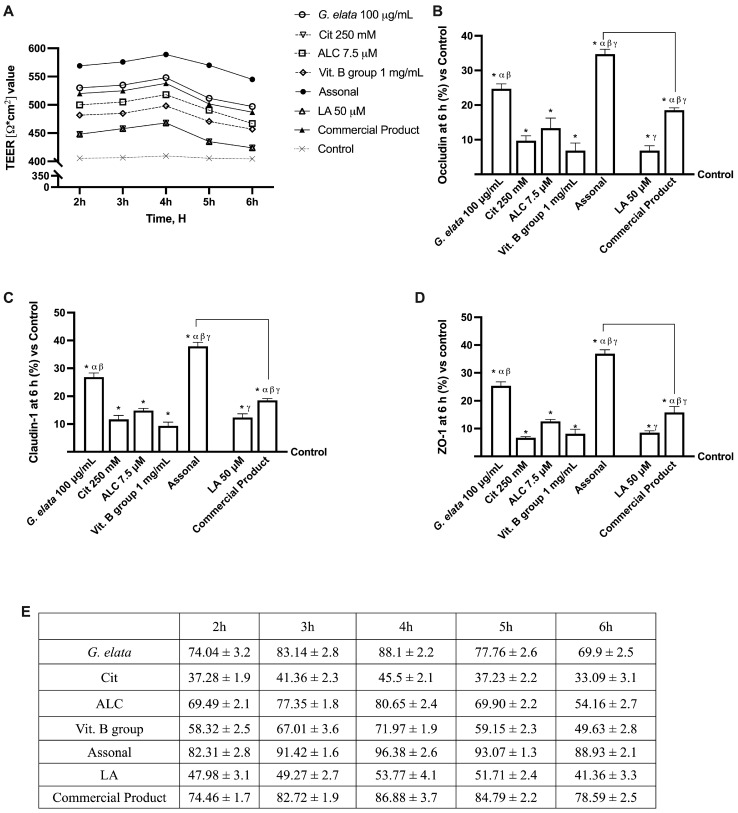
Permeability study on CaCo-2 cells. (**A**) TEER (transepithelial electrical resistance) Value using EVOM3 during the time (from 2 h to 6 h); (**B**–**D**) the analysis of TJ measured by Enzyme-Linked Immunosorbent Assay (ELISA) test (Occludin, Claudin-1, and ZO-1, respectively) at 6 h; (**E**) the absorption rate during time (from 2 h to 6 h) obtained from the application of following formula: J = Jmax [C]/(Kt + [C]). The abbreviations are the same as reported in Figure 2. In (**A**), data are expressed as means ± SD of five independent experiments. From (**B**–**D**), means ± SD are expressed comparing data to the control (0% line) of five independent experiments performed in triplicates, and all molecules result in * *p* < 0.05 vs. control; α *p* < 0.05 vs. Cit, ALC, and Vit. B group; β vs. LA; γ vs. *G. elata*; the bars *p* < 0.05 vs. commercial product.

**Figure 4 ijms-25-02376-f004:**
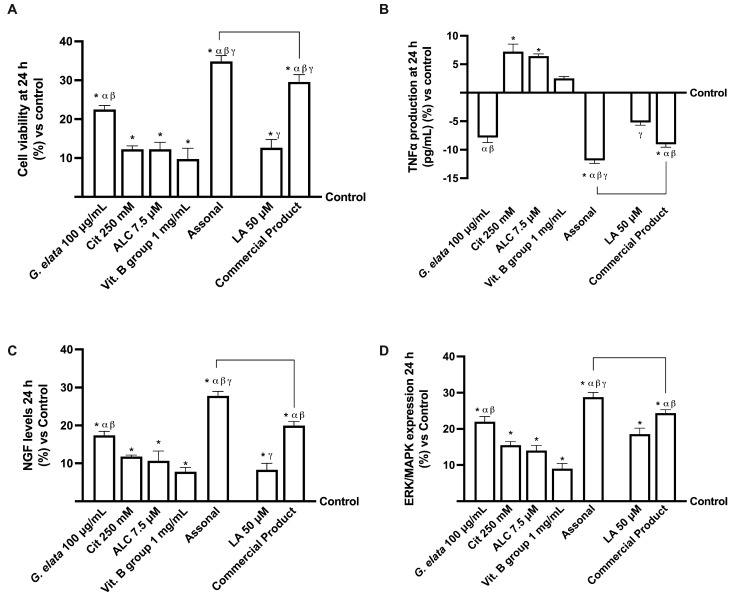
Analysis of 3D EngNT in physiological conditions. In (**A**), cell viability is measured by the MTT test; (**B**), TNFα, (**C**), NGF, and (**D**), pERK/MAPK are measured by the ELISA kit. The data obtained at 24 h of stimulation are expressed as mean ± SD of five independent experiments performed in triplicate and normalised to control values (0% line). The abbreviations are the same as reported in Figure 2. * *p* < 0.05 vs. control; α *p* < 0.05 vs. Cit, ALC, and Vit. B group; β vs. LA; γ vs. *G. elata*; the bars *p* < 0.05 vs. commercial product.

**Figure 5 ijms-25-02376-f005:**
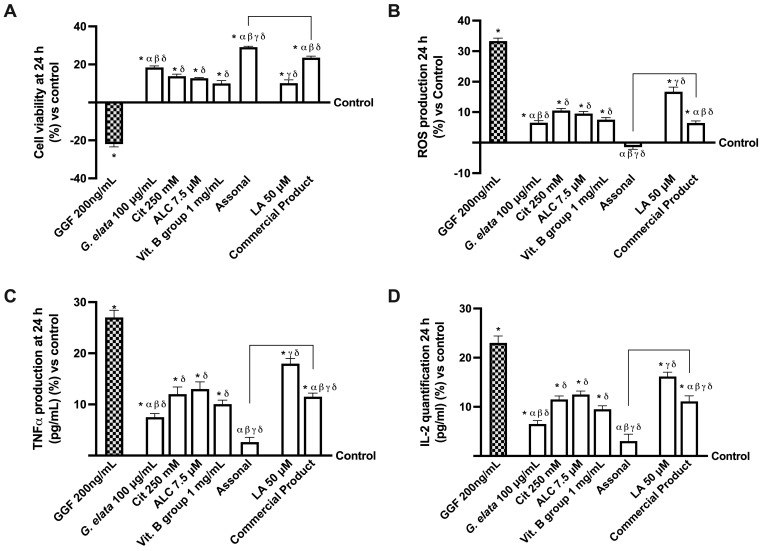
Analysis of single agents and combinations effects under PNI conditions at 24 h of stimulation. In (**A**), cell viability by MTT; in (**B**), analysis of ROS production measured by cytochrome C reduction; in (**C**), TNFα quantification by ELISA test; and in (**D**), IL-2 analysis by ELISA test. The data are expressed as means ± SD (%) of five independent experiments performed in triplicate and normalised to control values (0% line). GGF = glial growth factor 2; the other abbreviations are the same as reported in Figure 2. * *p* < 0.05 vs. control; α *p* < 0.05 vs. Cit, ALC, and Vit. B group; β vs. LA; γ vs. *G. elata*; δ *p* < 0.05 vs. GGF 200 ng/mL; the bars *p* < 0.05 vs. commercial product.

**Figure 6 ijms-25-02376-f006:**
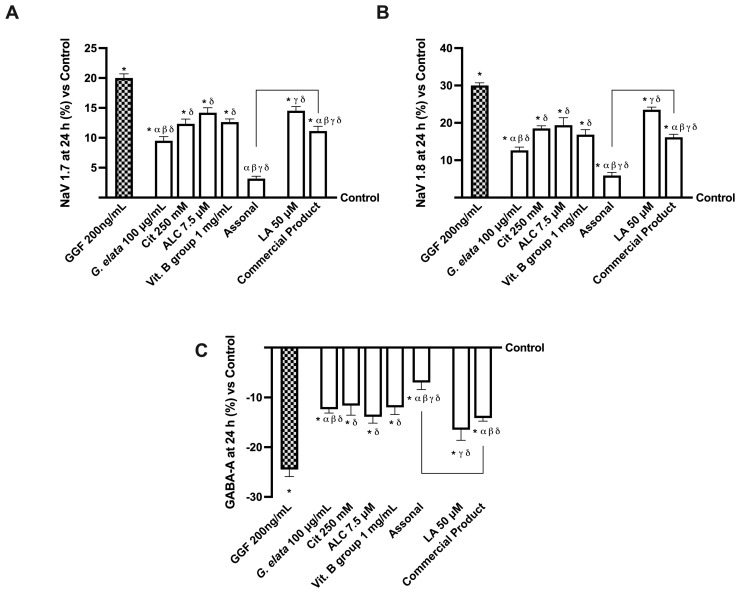
Analysis of single agents and combinations on NaV channels and GABA level under PNI conditions at 24 h of stimulation. In (**A**), Nav 1.7; in (**B**), Nav 1.8; in (**C**), GABA-A. All these tests were performed with a specific ELISA kit. The data are expressed as means ± SD (%) of five independent experiments performed in triplicate and normalised to control values (0% line). The abbreviations are the same as reported in Figure 5. * *p* < 0.05 vs. control; α *p* < 0.05 vs. Cit, ALC, and Vit. B group; β vs. lipoic acid; γ vs. *G. elata*; δ *p* < 0.05 vs. GGF 200 ng/mL; the bars *p* < 0.05 vs. commercial product.

**Figure 7 ijms-25-02376-f007:**
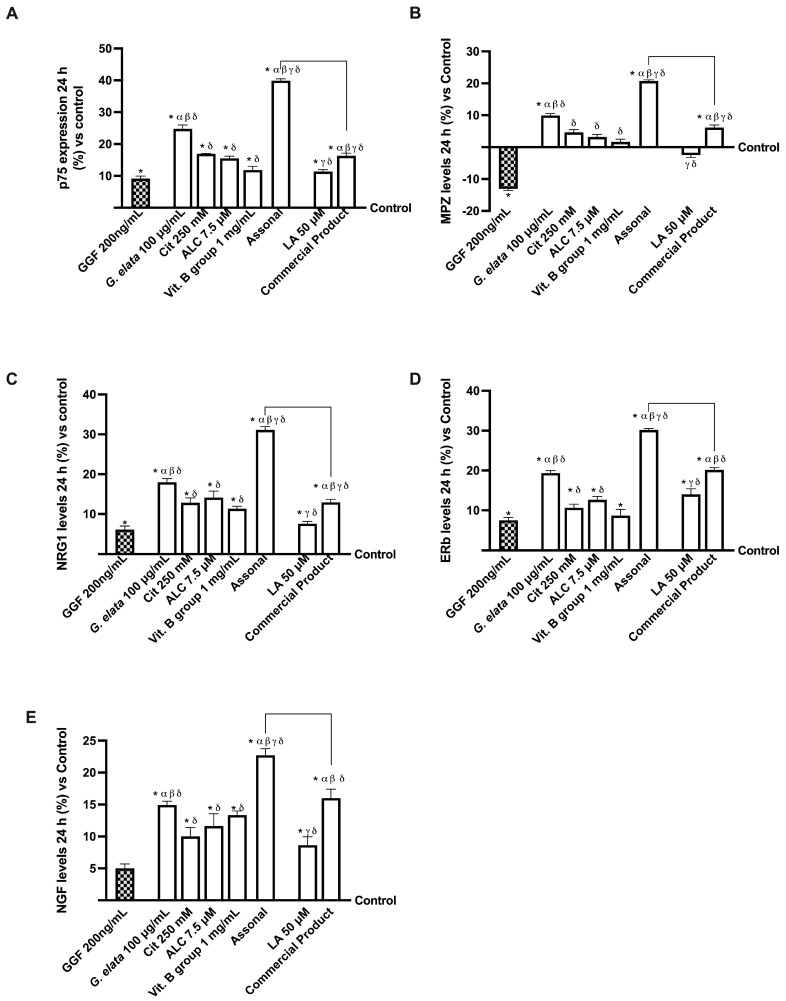
Analysis of single agents and combinations on PNS molecular marker under PNI conditions at 24 h stimulation. In (**A**), p75 expression by ELISA test; in (**B**), myelin protein zero (MPZ) by ELISA test; in (**C**), NRG1 by ELISA test; in (**D**), ERb by ELISA test; in (**E**), NGF level measured by the ELISA kit. The data are expressed as means ± SD (%) of five independent experiments performed in triplicate and normalised to control values (0% line). The abbreviations are the same as reported in Figure 5. * *p* < 0.05 vs. control; α *p* < 0.05 vs. ALC and Vit. B group; β vs. LA; γ vs. *G. elata*; δ *p* < 0.05 vs. GGF 200 ng/mL; the bars *p* < 0.05 vs. commercial product.

**Figure 8 ijms-25-02376-f008:**
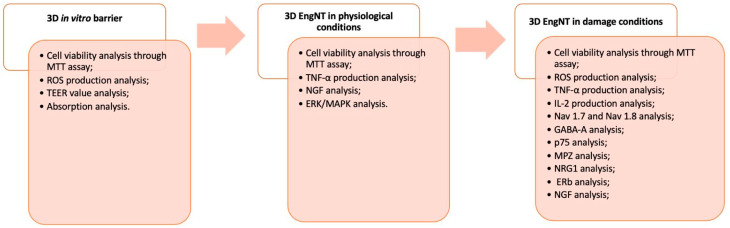
Diagram of the experimental protocol divided into 3 different phases.

## Data Availability

Raw data are deposited at the Laboratory of Physiology (C. Molinari), ensuring appropriate measures so that raw data are retained in full forever under a secure system. The data presented in this study are available on reasonable request from the corresponding author.

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
