# Peer review of "Effects of Nutraceutical Compositions Containing Rhizoma Gastrodiae or Lipoic Acid in an In Vitro Induced Neuropathic Pain Model"

_ijms, 2024, doi:10.3390/ijms25042376_

Round 1

Reviewer 1 Report

Comments and Suggestions for Authors

The paper of Ferrari et al. is a very good example of an experimental research conducted on food supplements made of a combination of natural products. The validation of "nutraceuticals" is crucial in the advancement of the sector of products for human health different from conventional drugs.

I truly appreciated the large in vitro protocol that authors used and I believe that results are worthy to be considered. I find that targets considered are very interesting and allow to discuss the effect of the combination in a proper way. A very important pro, moreover, is the use of a positive control.

Here below my major and minor concerns, reported as suggestion to improve the quality of the work.

Major concerns:

- I can not understand if the used sample is gastrodin as pure compound or Gastrodia elata extract. Authors should fix this point.

If pure gastrodin is used, the reference to the extract is misleading; if the extract is used, authors should provide a chemical characterization and change w/w measure in microg/mL not in microM.

- I suggest to normalize values at 100%, not 0%, because the visualization of values will be clearer and differences more marked.

- The statistical analysis is not very clear, becasuse authors reported a triplicate of five experiments (is it correct?), but SD is very, too much, restrained and in some cases, such as fig. 2 or table 3, is missing.

- I suggest authors to clearly indicate time treatments for all different experiments.

- In ERK dosage, authors dosed p-ERK or total ERK?

Minor concerns:

- I suggest to use the term "expression" and not activity as regards ELISA dosages.

- I suggest to use the complete botanical name Gastrodia elata Blume for the first time, then: G. elata; ok for Rhizoma Gastrodiae. I also suggest to write the name of molecules in lowercase 

Comments on the Quality of English Language

The English language requires moderate revisions

Author Response

Best regards

Francesca Uberti

Reviewer 2 Report

Comments and Suggestions for Authors

The manuscript entitled 'Evaluation of the protective effect of the bioactive compound of Rhizoma Gastrodiae compared to lipoic acid in an in vitro induced neuropathic pain model.' is an important article of great interest to scientists and readers in the field of peripheral neuropathy and peripheral nerve regeneration. The authors put a great effort into analyzing with different approaches the beneficial activity of the Rhizoma Gastrodiae-derived phenolic glycoside Gastrodin and the combination of it with a cocktail of bioactive molecules, that they named Assonal. The results obtained are neatly in favor of the Assonal compound and hold promise, together with several other experimentally active strategies, to be a potential therapeutic tool of clinical relevance.

However, the significance of the research being of high topical interest and to increase the scientific impact of the paper, there are some remarks concerning mainly the quality of the presentation,  the overall presentation of the results and discussion, and the description of methods.

Results: they are very complete and rich in several assays, details, and molecules. For this reason, even if the text has been divided into paragraphs/subheadings, the reader can be at a loss without a guided plan of the experiments. Perhaps, a table or a diagram, or table + diagram telling and resuming the several experimental tests can be of help.

Moreover, in some phrases, authors comment on the results as if they were discussing their significance, and this is helpful on one side, but on the other can be confusing is very confusing. The same happens in the discussion, where some data belonging to the results are reported to add significance to the comments.

Examples: line 164 'indicating the possibility of a synergistic effect using all substances in the same formulation.'; lines 204-207: 'All these results revealed the combination....better absorption profile than a commercial product, confirming an improvement in the bioavailability and suggesting a synergistic effect of the combination'

Discussion: the text is difficult to follow and, though complete in every aspect regarding the presented research,  dividing the text into paragraphs could certainly help the clarity and improve the readability.

Materials and Methods: 

Some paragraphs need to be subdivided with subheadings, others need to be grouped.

Paragraph 4.2: it could be helpful to subdivide into 4.2.1, 4..2.2, and 4.2.3 with subheadings dedicated to using the cell lines. Could paragraph 4.8 be included in this as subheading 4.2.4? 

Paragraph 4.3: a Gantt diagram or similar should accompany the description. Possibly, the next paragraphs (4.4., 4.5. 4.6, 4.7), should be grouped under the 4.3 one.

Paragraphs 4.9-4.20: they are all descriptions of Elisa Assays and should be grouped under a leading paragraph dedicated to the ELISA determinations and relative subheadings with the details of examined molecules.

Conclusions: The first sentence dedicated to the Gastrodin alone is not representative of the presented results and should be removed.

Minor remarks:

Title:  the title is a standalone phrase and does not need the full stop at the end 

NRG1 and GGF: in the text, the difference and significance between the two. 

Comments on the Quality of English Language

Please revise sentence construction and style, as well as phrase lenght.  

Author Response

Best regards

Francesca

Reviewer 3 Report

Comments and Suggestions for Authors

This manuscript explores the beneficial effects of gastrodin (the main component of Gastrodiae Elata extract) as an alternative to lipoic acid in nonpharmacological  antioxidant treatments for peripheral neuropathic pain. 

In both cases, the treatment is carried out by a combination of several agents, and this would be clearly stated at the title. Gastrodin or lipoic acid are one of the components of the combination with other common antioxidant and nootropic agents (citicholine, N-Acetylcarnitine and a mixture of different vitamins B). It is not a comparison of just gastrodin vs lipoic acid. The combination should be clearly mentioned in the title and the abstract.

The mixtures, named Assonal (with gastrodin) or “a commercial product” (with lipoic acid)  should be mentioned at the introduction for a better comprehension of the study. The name of the commercial product would be also included.

Major concerns

Lipoic acid is a cofactor of intramitochondrial pyruvate dehydrogenase and other mitochondrial alpha-keto acid dehydrogenases. This is the reason why the supply of lipoic acid is an activator of the mitochondrial respiration. However, gastrodin (HBA after glycosylase action on gastrodin, which is a glycon form) is not needed for those enzymes, and it is not directly involved in mitochondrial respiration, but it is an antioxidant that directly decreases ROS by scavenging those species probably activating enzymes of the Nrf2-mediated antioxidant responses. In other words, the mechanisms of action of lipoic acid and HBA are different, and this should be considered and discussed.

Methods

It is not demonstrated that the ratio among the concentrations of the four components, and vitamin B types would be the same in Assonal and the "commercial product". As soon as the control for comparison is a commercial product its composition should be given.

Other points to be addressed before acceptance.

Line 42-43:  The sentence “researchers' attention has now shifted to the role of non-neuronal cells. I suggest that the authors should modify the sentence. They mean peripheral neurons, or neurons out of Central Nervous System, but they are neurons yet. Non-neuronal cells are not correct.

Line 167: Acetyl-L-Acetyl-L-Carnitine. I think that the component is Acetyl-Lcarnitine. Double acetylation in very unlikely. In addition, Legend of Figure 1 is referred as Acetyl-L-carnitine. Please, check and repair the name of this component throughout the manuscript  (lines 381, 458, 464, 468 and other locations).

Figure 3E. Standard deviation should be included at the Table.

Line  578: Km of what? What enzyme is involved? Is fluorescein a substrate of the unknown enzyme?

Line 697: Define MPZ and NRG1 at the abbreviation list.

Author Response

Best regards

Francesca

Round 2

Reviewer 1 Report

Comments and Suggestions for Authors

Authors fixed all major and minor concerns.

Reviewer 2 Report

Comments and Suggestions for Authors

The revised version of the manuscript has neatly improved compared to the original and I hope the authors got the same belief. 

After reading the rebuttal letter, I could see that the authors accomplished all the suggested improvements. I also appreciate the changed title, even though was not a remark of mine, since now is more representative of the research content. The authors also included the diagrams of experiments and results and they clearly illustrate the take-home message. 

Reviewer 3 Report

Comments and Suggestions for Authors

I have read with attention the reply letter and the modified version of the manuscript, and I would appreciate all the replies and comments supplied by the authors. I think that the manuscript has been improved and I hope authors think the same. Manuscript improvement is the aim of the peer review process.

 According to that, I think that all the modifications satisfy my concerns. I think that the title and aim of the study are clear, as well as the composition of the nutraceutical. The brief paragraph concerning the different mechanisms of action is helpful and appropriate, although I understand that this is not the main goal of the study.

 Finally, I would like to write a comment about the Kt at the legend of Figure 3. I apologize since I did a mistake concerning Km instead of Kt in my review.v1. I know the differences both. Kt is the same concept but referred to transport instead of enzymatic catalysis. So, there is no enzyme involved, but a transport (protein or system). So, it is ok and that concern was my fault. I am sorry.